# Can Swallowing Cerebral Neurophysiology Be Evaluated during Ecological Food Intake Conditions? A Systematic Literature Review

**DOI:** 10.3390/jcm11185480

**Published:** 2022-09-18

**Authors:** Yohan Gallois, Fabrice Neveu, Muriel Gabas, Xavier Cormary, Pascal Gaillard, Eric Verin, Renée Speyer, Virginie Woisard

**Affiliations:** 1Laboratory LNPL—UR4156, University of Toulouse-Jean Jaurès, 31058 Toulouse, France; 2ENT, Otoneurology and Pediatric ENT Department, Pierre Paul Riquet Hospital, University Hospital of Toulouse, 31059 Toulouse, France; 3Independent Researcher, Swallis Medical, 31770 Colomiers, France; 4Laboratory CERTOP—UMR CNRS 5044, Maison de la Recherche, University of Toulouse-Jean Jaurès, 31058 Toulouse, France; 5Independent Researcher, 31600 Muret, France; 6Laboratory CLLE CNRS UMR5263, University of Toulouse-Jean Jaurès, 31058 Toulouse, France; 7Department of Physical and Rehabilitation Medicine, Rouen University Hospital, 76000 Rouen, France; 8Department Special Needs Education, University of Oslo, 0318 Oslo, Norway; 9Curtin School of Allied Health, Faculty of Health Sciences, Curtin University, Perth, WA 6102, Australia; 10Department of Otorhinolaryngology and Head and Neck Surgery, Leiden University Medical Centre, 2333 ZA Leiden, The Netherlands; 11Voice and Deglutition Unit, Department of Otorhinolaryngology and Head and Neck Surgery, Larrey Hospital, University Hospital of Toulouse, 31059 Toulouse, France

**Keywords:** cerebral activity, swallowing function, fNIRS, magnetoencephalography, electroencephalography

## Abstract

Swallowing is a complex function that relies on both brainstem and cerebral control. Cerebral neurofunctional evaluations are mostly based on functional magnetic resonance imaging (fMRI) and positron emission tomography (PET), performed with the individual laying down; which is a non-ecological/non-natural position for swallowing. According to the PRISMA guidelines, a review of the non-invasive non-radiating neurofunctional tools, other than fMRI and PET, was conducted to explore the cerebral activity in swallowing during natural food intake, in accordance with the PRISMA guidelines. Using Embase and PubMed, we included human studies focusing on neurofunctional imaging during an ecologic swallowing task. From 5948 unique records, we retained 43 original articles, reporting on three different techniques: electroencephalography (EEG), magnetoencephalography (MEG) and functional near infra-red spectroscopy (fNIRS). During swallowing, all three techniques showed activity of the pericentral cortex. Variations were associated with the modality of the swallowing process (volitional or non-volitional) and the substance used (mostly water and saliva). All techniques have been used in both healthy and pathological conditions to explore the precise time course, localization or network structure of the swallowing cerebral activity, sometimes even more precisely than fMRI. EEG and MEG are the most advanced and mastered techniques but fNIRS is the most ready-to-use and the most therapeutically promising. Ongoing development of these techniques will support and improve our future understanding of the cerebral control of swallowing.

## 1. Introduction

The swallowing function is critical to ensure survival in all species, allowing both nutrition and airway protection. Its volitional and spontaneous coordination involves about 30 nerves and muscles [1], through three phases: the oral phase, the pharyngeal phase and the esophageal phase. Each phase has its own neuromuscular structures with specific brainstem control centers. The peripheral motor function has been well-described by many authors through multiple techniques such as flexible endoscopy of swallowing, videofluoroscopy, pharyngo-esophageal manometry and cervical electromyography. Its brainstem reflexive regulation is also well-described in anatomical and neurofunctional studies [2,3]. Anatomical studies suggest that a swallowing central pattern generator is located in the nucleus ambiguus and retroambiguus in the medulla [1,4]. 

However, less is known about the central cortical regulation of swallowing. For many years, swallowing was thought to be a reflex action, it was believed to involve solely the brainstem. It was Hamdy et al.’s work using transcranial magnetic stimulation (TMS) and functional magnetic resonance imaging (fMRI) that suggested otherwise, resulting in TMS and fMRI being the new gold standard for neurofunctional explorations [5,6]. 

For instance, with fMRI, Martin et al. compared the cortical activation associated with the swallowing function and non-swallowing tongue motility [7]. They showed specific swallowing activation located in the lateral post central gyrus, the supra marginal gyrus, the cuneus and precuneus. Next, Luan et al. explored the connectivity of the complex swallowing network with fMRI, showing it to be bilateral and symmetrical [8].

Dysphagia is the symptomatic expression of swallowing difficulties. Its pathophysiological mechanisms are numerous, in accordance with the numerous structures involved in the swallowing function. Dysphagia can result from many different underlying diseases affecting either the muscles involved in swallowing or the neurological control of swallowing.

At the neurofunctional level, the results from previous neurological studies have limited value in the daily clinical evaluation of dysphagia and may not be easily generalizable from one subject to another, particularly in case of neurologic disease (dystonia) or in children. Indeed, MRI is not widely available, and, above all, the subject must be lying down, which is not a natural position for meals in adults. Moreover, motion artifacts highly impair fMRI results; thus, a prolonged immobility is required for good analysis. Therefore, subjects presenting with dystonia, uncontrolled movement or unable to lay down properly due to neurologic disease and children are not easy populations to perform fMRI, even if they would be of interest. Another neurofunctional imaging method, the positron emitting tomography (PET), shares similar drawbacks, in addition to its radiating nature that increases the risk of neoplastic diseases. Electrocorticography being an invasive technique, requiring a surgical placement over the cortex, it is also not adapted for the aforementioned situations and is reserved for patients undergoing cranial surgery. It appears that non-invasive non-radiating explorations of the cerebral activity would be of particular interest in the study of the swallowing function. 

The present study aims to systematically review non-invasive non-radiating neurofunctional tools, other than fMRI and PET, that can be used to explore the cerebral activity during a food swallowing task occurring in a natural setting. Furthermore, we will describe their results in the light of possible applications in routine clinical assessments. 

## 2. Materials and Methods

### 2.1. Protocol and Registration

We performed this literature review according to the preferred reporting items for systematic reviews and meta-analyses (PRISMA) statement [9]. The PRISMA statement guides researchers in their reports of systematic reviews. This protocol was registered on Prospero as ID 319150 (University of York, York, UK).

### 2.2. Data Sources and Search Strategies

We performed systematic literature searches from Embase (Elsevier, Amsterdam, Netherlands) and PubMed (MEDLINE, Bethesda, MD, USA). All publications prior to 30 September 2021 were included, with no limitations regarding publication dates. The complete search strategies including both subject headings (e.g., MeSH and thesaurus) and free text terms are presented in Table 1.

Four reviewers participated in the two-step inclusion process. For every article, two reviewers filled each of the eligibility criteria and an excel function permitted to include/exclude each article according to these criteria to prevent errors of selection (e.g., inclusion in spite of the presence of an exclusion criteria). In case of divergent opinion, the reviewers discussed the article until they found a consensus. The selection process flowchart is shown in Figure 1.

### 2.3. Eligibility Criteria

The first step of screening was based on titles and abstract reviewing. Inclusion criteria for this step were the following: studies reported on (1) swallowing function and/or dysphagia; (2) brain structures, (3) brain signals (4) with an “ecological” swallowing situation. “Ecological” refers to any food intake situation in a usual/natural position (seated or standing up) with no invasive evaluation method. Invasive (e.g., intracerebral electrodes) and/or radiating techniques (e.g., PET), inappropriate head or body positioning (e.g., laying down for MRI) were considered “non-ecological”. Studies using these methods, without any other eligible task or technique, were thus excluded. Only articles published in English or French were eligible. 

Case reports, animal studies, reviews, congress papers, post-mortem or fetal studies and articles involving a technique that was clearly “non-ecological” were excluded. 

The second step of inclusion was based on the full article reading. Inclusion criteria were slightly refined with the following additional criteria: (1) during task the subjects were requested to swallow and/or specifically focus on their swallow; (2) awake and alert during task (excludes swallowing during sleep); (3) the meal position had to be stated (whether sitting or standing up); (4) the brain signal had to be clearly recorded during the swallowing task.

### 2.4. Methodological Quality, Level of Evidence, and Risk of Bias

Methodological quality assessment was rated by two independent researchers, after which a consensus was reached with the involvement of a third reviewer, when necessary. Two tools were used to assess the quality of the included articles. To assess their level of evidence, we used the National Health and Medical Research Council (NHMRC) Evidence Hierarchy, ranging from I (Systematic Review) to IV (Case Serie) [10]. To assess their methodological quality, we used the QualSyst critical appraisal tool published by Kmet et al. [11]. The QualSyst tool is a systematic and replicable assessment tool evaluating the methodological quality of a broad range of study designs. The Qualsyst has a three-point ordinal scoring system (yes = 2, partial = 1, and no = 0). The total Qualsyst score can be converted into a percentage score: strong quality for scores from 80 to 100%; good quality for 60 to 79%; adequate quality for 50 to 59%; and poor quality below 50% scores. 

### 2.5. Data Extraction and Synthesis of the Results

Once studies were included and their quality was evaluated as sufficient (Qualsyst score above 50%), data across all studies was extracted. To achieve this, comprehensive data extraction forms were used, focusing on: the number of subjects and subjects characteristics (healthy versus dysphagic, their age, gender and hand lateralization); the neurofunctional method used, the number of channels/captors and cerebral coverage; the studied outcome parameters (type of tasks, bolus types and period of recording, e.g., swallowing preparation and/or execution) and statical analysis methods; and lastly, the authors principal conclusions. Geographic bibliometric data were also recorded.

We grouped the studies according to the technology used, as analysis methods vary according to the type of signal recorded. Motor imagery and direct pharyngo-esophageal stimulation without swallowing execution were not considered as part of the focus of this review. Results of the inclusions are presented below. 

Based on studied population and performed tasks, we classified the studies as referring to “physiology” (classical swallowing condition in healthy subjects), “adaptive physiology” (modification of a specific characteristic of interest during the task in healthy subjects, such as body and head positioning or sensory stimulation), “pathology” (comparison between dysphagic population and healthy subjects) and “patho-physiologic” (if the study would fall under both “pathology” and “adaptive-physiology”).

Two triggering methods for the swallowing task were differentiated: non-volitional and volitional tasks. The non-volitional tasks, included both “provoked swallowing” (shunt of the oral phase) and “spontaneous swallowing”, whilst the volitional tasks, included both “cued swallowing” (investigator-driven or informatically driven) and “self-paced swallowing” (subject-driven).

### 2.6. Data Presentation and Analysis

The descriptive data is presented at a global level and per technique. When possible, we described numeric data using means, standard deviations and range (Mean ± SD; Range), unless specified otherwise. 

## 3. Results

### 3.1. Study Selection

A total of 5948 unique records were retrieved. Each of the four reviewers assessed 50% of all records. After title and abstract selection, 208 full-text articles were screened for eligibility, of those, 43 articles were included in this review (Figure 1). The list of the included articles and their methodological quality as evaluated by QualSyst are presented in Table 2. Their respective results are available in Appendix A.

### 3.2. Neurofunctional Imaging Techniques

We identified three techniques used in neurofunctional swallowing assessment that were both non-invasive, non-radiating and usable in standard meal positions: electroencephalography (EEG), magnetoencephalography (MEG) and functional Near Infrared Spectroscopy (fNIRS). A total of 21 (50.0%) articles focused on MEG, 11 (26.2%) on fNIRS and 10 (23.8%) on EEG. Table 3 summarizes the complete Appendix A as a presentation of each method’s technical characteristics in comparison to fMRI, the Gold Standard of functional neuroimaging. A detailed report on the contributions provided by each technique, with regards to the cerebral activity during swallowing, will be provided. The results will be presented according to the methods, analyses and limitations of each technique. 

### 3.3. Quality Assessment

According to the QualSyst score [11], 36 studies displayed a strong methodology (EEG: 9; MEG: 19; fNIRS: 8) and the remaining 7 had good methodology (EEG: 2; MEG: 2; fNIRS: 3).

Based on NHMRC evidence hierarchy [10], 1 MEG study was classified as level II (“Randomized control trial”), 2 MEG studies as level III-1 (“Pseudo randomized control trial”); 4 studies (EEG: 2; fNIRS: 2) were classified as level III-2 (“Comparative studies with concurrent controls and allocation not randomized [cohort studies], or case control studies”); 9 studies (MEG: 7; fNIRS: 2) were classified as III-3 (“Comparative studies with two or more single-arm studies”); 27 studies (EEG: 9; MEG: 11; fNIRS: 7) were classified as level IV (“Case series”). The full quality evaluation is displayed in Table 2.

Since no studies were found to be of poor quality, all 43 eligible studies were included. On the whole, all the studies selected were assessed as being of good quality; however, the majority of studies displayed a relatively low level of evidence (mostly level III and IV, see Table 2).

### 3.4. Bibliometric Data

Regardless of the technique used, multiple teams are involved in natural neurofunctional evaluation of swallowing, mainly in Europe, North America and Asia. We found 6 international collaborations. EEG is most used in the USA (N = 6) and Japan (N = 3), and we noted 1 collaboration between USA and Japan, and 1 between USA and Austria. MEG is the most used technique, mostly in Germany (N = 18). Germany had 2 collaborations, 1 with the United Kingdom and 1 with Canada. fNIRS has been the most experimented in Austria (N = 5) and Japan (N = 4). One collaboration between Austria and Germany was noted. The worldwide study distribution is displayed in Figure 2.

EEG and MEG studies have been published since the 2000s, while most of fNIRS studies date from the mid 2010s as fNIRS is a younger technology (Figure 3).

### 3.5. Participants

Amongst the 43 studies, the mean number of participants was 17.5 ± 13.4, ranging from: 3 to 55 (EEG: 24.8 ± 20.1; Range= 7–55; MEG: 14.2 ± 9.2; Range= 3–44; fNIRS: 16.5 ± 9.8; Range = 6–43). Only 10 studies included 20 subjects or more (23.3%) [16,17,18,19,35,36,37,38,53], 31 studies included between 5 and 19 subjects (72.1%) [12,13,14,15,20,21,22,25,26,27,28,29,30,31,32,33,34,39,40,41,42,43,44,45,46,47,48,49,50,51,52], and the 2 studies that had less than 5 subjects (4.7%) [23,24] were the first published on this topic. 

A total of 32 (74.4%) studies focused on healthy subjects [12,13,14,15,16,17,18,19,21,23,24,25,26,27,28,29,31,32,34,38,40,42,43,44,46,47,48,49,50,51,53,54], 10 studies (23.3%) included both healthy and dysphagic subjects [20,22,30,33,35,36,37,39,45,52] and 1 study (2.3%) focused on only dysphagic subjects [41]. Appendix A displays these data per technique. 

Only one EEG study focused on children aged from 10 to 13 years [22]. All other studies focused on adult subjects, aged between 18 and 80 years. More details can be seen in Appendix A.

Regarding gender distribution, 7 (16.3%) studies did not display this data [14,17,18,24,27,33,41] and 1 study (2.3%) seemed to present a mistake describing “14 females and 7 females” [54]. Among the other 35 (81.4%) studies, gender distribution was slightly unbalanced, with a total of 268 females (F) and 305 males (M), with notably more women in fNIRS studies (97F/77M) and more men in MEG studies (101F/153M). 

Hand lateralization can affect cortical activity distribution and should be recorded in studies on cortical activity. However, 17 studies (39.5%) did not present this data [16,17,18,21,22,30,33,37,39,41,42,43,46,47,50,52,53]. The 26 (60.5%) other studies focused mainly on right-handed subjects. Throughout these studies, only three ambidextrous subjects from three different studies [19,35,36] and one left-handed subject [35] were included. Unfortunately, their individual results were not specifically described. Therefore, further conclusions might be applicable only to right-handed subjects.

A quarter of studies (11, 25.6%) explored dysphagic subjects, either alone (n = 1; 2.3%) [41] or compared to healthy controls (n = 10; 23.3%) [20,22,30,33,35,36,37,39,45,52]. Looking at dysphagic subjects in more detail, they suffered from different diseases and conditions, including: stroke [20,35,45,52], Parkinson’s disease [37], amyotrophic lateral sclerosis [36], Kennedy disease [33], botulism [30], anterior open bite [22] and functional dysphagia [39]. 

### 3.6. Types of Studies

The study types were the following: 17 (40%) “physiology” studies (5 EEG, 7 MEG, 5 fNIRS); 15 (35%) “adaptive physiology” studies (4 EEG, 7 MEG, 4 fNIRS), 2 (5%) “patho-physiology” (1 EEG, 1 fNIRS), and 9 (21%) “pathology” studies (1 EEG, 1 fNIRS, 7 MEG). More specifically, “adaptive physiology” focused on the effects of different conditions: differences in age [34,53], bolus viscosity [16,18], head positioning [17,18], bolus volume [21], distractors [21], taste [48], tactile stimulations [29,31,40,42,43,49], neurofeedback [46] and transcranial direct current stimulation (tDCS) [38,42]. The 2 “patho-physiology” studies focused on the effect of tDCS [20] and the taste perception differences [52] between stroke patients and healthy subjects.

The studies looked at different phases of the swallow A total of 6 (14%) studies focused specifically on swallowing preparation (4 EEG, 2 MEG), 15 (35%) specifically on execution (1 EEG, 3 MEG, 11 fNIRS), and 22 (51%) focused on both (6 EEG, 16 MEG).

When studying the cortical activity of swallowing, the different types of boluses evaluated are of importance. The majority of studies evaluated swallowing with liquids (29 studies = 67%), or saliva (8 studies = 19%), or both (3 studies = 7%). The remaining study (2%) focused on solids through the swallowing of cucumber. The swallowing of liquids was mostly assessed with water in all but one study. The effect of viscosity on the swallow was also reported in 2 studies [16,18], using honey or nectar. Furthermore, one study used flavored jellies [52]. It is of note that taste, more than swallowing itself, was the major point of interest in two studies using water [43] or jelly [52].

The type of swallowing trigger can also influence the observed cortical activity. In terms of triggered swallow, 34 studies (79%) focused on self-paced, 7 studies (16%) on cued, 2 studies (5%) on spontaneous and 2 studies (5%) on provoked swallowing. Three studies compared different triggers: self-paced versus cued swallowing [15], cued versus provoked swallowing [20] and self-paced versus provoked swallowing [25].

See Table 4, Table 5 and Table 6 and their respective extended versions, Appendix A for each study details.

## 4. Discussion

The present review highlighted three different neuro-imaging techniques currently used for swallowing assessment purposes that meet our ecological process of swallowing. As we focused on the study of motor activity in an ecological swallow, other tasks such as motor imagery, direct pharyngo-esophageal stimulation and tongue movement without swallowing studies results were not included.

For each technique, it is important to understand how the cerebral area can be accessed in order to better understand their specific results. They share some similarities in terms of technical procedures, objectives and results but also have their own specificities, which explain their complementary nature. 

### 4.1. Methodological Considerations

All three techniques can efficiently analyze the superficial cortical activity, but for deeper activity, fNIRS is limited and EEG or MEG are necessary. On the one hand, fNIRS is considered as a **topographical** (it is limited to the surface of the cortex, resembling a surface map) superficial technique since its signal is limited to the cortical area 3 cm below the scalp skin, which matches the most superficial part of the cortex. On the other hand, MEG and EEG are **tomographical** techniques that can render three-dimensional results. Those two are supposedly able to access both superficial cortices and deep brain regions, such as insula and deep grey nuclei, thanks to specific statistics analyses (e.g., independent component analysis, graph theory and/or wavelet analyses). 

With this in mind, we will focus first on the regions of interest for this review; second, the variations according to different task types and bolus substances and lastly, the studies will be differentiated based on their physiologic and/or pathologic objectives and results.

#### 4.1.1. Activity Localization

It is important to delineate the main swallowing regions of interest from this review. On the one hand, all three techniques showed good activity of the caudal pericentral cortex (Appendix A), gathering both primary sensory and motor cortices (Broadman Areas (BA) 4,1,2,3). These regions likely focus on pharyngeal-laryngeal cortices according to both classical homunculus [59,60] and Ludlow et al.’s review [1]. Other areas can also be studied by both techniques, particularly in the premotor areas (premotor cortex [PMC], supplementary motor area [SMA] in BA6) and the prefrontal cortex (BA9, 10, 45) [20,33,39,47,52]. Over those regions, the three techniques allow for the study of both the precise localization of the activity and its chronology. These regions have also been described in fMRI studies and thus, are good candidates for further exploration in ecologic conditions [4,55]. 

On the other hand, some differences must be raised, as they are critical for the choice of the technique adapted to one’s experiment. The insula, the anterior cingulum cortex (ACC) and posterior cingulum cortex (PCC) have been shown to play an important role in the swallowing process in fMRI studies [4,55]. However, as a topographical superficial technique, the fNIRS is currently unable to provide good activity data of these deep regions. Whereas MEG results included in our studies, using tomographical source positions analyses, provided results consistent with fMRI results. No EEG studies mentioned these deeper activities, as they mostly focused on preparation potential without gyrus dipole source localization approach, or they focused on the network microstructure. Actually, only one study focused on the dipole source location of the activity and found an activity centered on the motor BA4 and PMC BA6 [19] and one study used the electrode as a topographical source locator [20], without three-dimensional dipole localization approach. We would still expect EEG to also be able to individualize this activity using independent component analysis. Indeed, other studies with classical somatosensory stimulation were able to individualize deeper activities such as the anterior cingulum [61]. 

Some spatial limitations of these techniques appear to be the putamen and the cerebellum activities shown in fMRI reports [4,55], as neither EEG, nor MEG (and obviously fNIRS) measured these activities in our review. This obstacle might not be crossable, but at least for the cerebellum, EEG and MEG could be of potential use according to Andersen et al. [62]. Similarly, fNIRS, EEG and MEG in their current technological state, appear impractical for exploring the brainstem and specifically the central pattern generator located in the nucleus ambiguus and retroambiguus. As mentioned previously regarding the cerebellum, EEG and MEG might be the future key explorations for this purpose. 

#### 4.1.2. Tasks Modalities

Secondly, in order to deepen each regions’ role, we need to separate different types of swallowing tasks, according to two characteristics that should be highlighted: the way to trigger the swallowing task and the bolus type, as each characteristic has its specificities. Through the review, task triggering was either non-volitional or volitional.

Non-volitional swallowing can be divided in two sub-tasks: provoked swallowing (direct pharyngeal instillation by external source) and spontaneous swallowing (the subject being unaware of the swallowing task). Two studies focused on provoked swallowing execution, with transnasal instilled water. In this regard, EEG showed an increase in approximate entropy (compared to resting state) of the central region in healthy subjects, just before the muscular swallowing activity [20]. MEG showed activation of the prefrontal cortex (BA9) during preparation (reflecting anticipation) and of the parietal lobule (BA7) during execution [25]. The transnasal canula sensation could be the cause of the parietal activity during the preparation phase, as suspected by the authors. Interestingly however, it was shown with MRI that tactile stimulation of the mucosa of the anterior nose elicit responses over the caudal part of the primary sensorimotor cortex, without parietal response [63]. To our knowledge, no study has tested the back of the nose/choanae cerebral sensory pattern. 

Spontaneous swallowing was the central point of two studies using fNIRS, focused on primary motor and primary sensory cortex. Without any instruction to the subject, they showed activity in the primary sensorimotor cortex [49,50]. It is of interest that the authors of those 2 fNIRS studies claim to be recording the BA4 but, according to the Bioimage suite tool (Yale School of Medicine, New Haven, CT, USA), might be recording the Broca Area (BA44). This detail might illustrate the lack of spatial resolution of current fNIRS system in comparison to EEG, MEG or fMRI, although its development is quickly improving. The fMRI was used in similar fashion (reported as “naïve saliva swallowing”) and showed similar activation of the lateral pericentral and PMC on both sides [64]. They also observed activation of the right insula, the superior temporal gyrus, the middle and inferior frontal gyri (MFG and IFG) and frontal operculum. The two fNIRS study channels were limited to the primary sensory and motor cortex, thus, they could not measure those activities, besides the deepness of certain areas. A broader cortical fNIRS coverage might confirm those fMRI results in the future.

Volitional swallowing can be cued swallowing (no decision from the subject) or self-paced swallowing (with the subject’s decision). For these tasks, studies focused either on the swallowing preparation and/or on the execution. It is also critical to take into account the swallowed substance, either saliva, liquids or solid. 

With regards to cued preparation of saliva swallowing, with EEG, Hiraoka et al. were able to measure cued evoked negative preparation potentials called CNV (“contingent negative variations”). CNV appeared earlier and were about 2 to 4 times stronger than the self-paced preparation potential named MRCP (“Movement related cortical potential” which contains the “Bereischaftpotential” [BP]). Those negative potentials were detected in the vertex area (Pz, Fz, Cz, C3, C4) [15]. Regarding water cued swallowing preparation, with EEG, Yuan et al. also showed a broader approximate entropy over the parieto-fronto-temporo-central region in cued condition in comparison to provoked conditions, where it is more limited to the central region [20]. With MEG, Watanabe et al. measured, during cued water swallowing, bilateral preparation responses that seemed to follow the following course, from the PCC, SMA, ACC, SFG, MFG, IFG and lastly, Insula, but only PCC and Insula had statistically significant different onset times (*p* < 0.003) [27].

During cued swallowing execution, EEG retrieved beta ERD in the motor and premotor regions (BA 4 and 6) with water in comparison to tongue tapping [19]. With fNIRS, an oxyHb increase can be measured in the whole pericentral, temporal and frontal regions associated to water swallowing [47], and also in SMC, PFC and pre-SMA while executing a swallowing task with chopsticks and cucumber slices [54]. This last task also evokes an oxyHb decrease in SMA and PMA after swallowing. This last study is the only to focus on the whole meal process, from the plate to the mouth. This illustrate its use in ecological conditions but could also be a source of confounding factors on its conclusions.

These results are consistent with fMRI results showing anterior cingulum, IFG, MFG, cuneus and precuneus region activation during volitional cued swallowing in comparison to spontaneous swallowing [64,65]. This cingulate activity appears to be linked with task complexity or with an imposed higher rate of swallowing [6]. With 18FDG-PET, Harris et al. showed broader activity of the left sensorimotor cortex, cerebellum, thalamus, precuneus, anterior insula, left and right lateral postcentral gyrus, and left and right occipital cortex [66].

Volitional self-paced swallowing preparation and execution were the most studied in the different articles. During preparation, all EEG studies tended to record a one-phase Bereischaftpotential (BP), a negative preparation potential recorded from the vertex region before self-paced motor tasks, about 1500 ms before swallowing muscular activity [12,13,14,15]. This BP appeared to be stronger with saliva than with water [14]. With MEG and water, a similar activity was shown to be located in the Cingulate gyrus and SMA (thought to be the origin of the BP), but also in MFG, IFG and insula with similar time onset (from −1500 ms to 1000 ms before muscular activity) [24].

Regarding the execution, EEG and MEG studies confirmed that the majority of the activity is localized in the pericentral cortex in the alpha and beta bands of the recorded signal. The evoked potential in an EEG study of saliva self-paced swallowing execution showed greater amplitude than water in Hiraoka et al.’s study [14]. Self-paced swallowing execution MEG studies represent a major part of our review, as it gathers all the work of Dziewas et al., which deserves a focus as it is the most advanced in term of both localization, temporal course of the activity and the implications of pathophysiological alteration on the swallowing network. They showed that in normal conditions, the alpha and beta activities are focused on the pericentral cortex and globally symmetric, but more precisely, they undergo a shift from the left to the right hemisphere from 400 ms before to 600 ms after the muscular activity onset, suggesting a left lateralization of the voluntary phases (oral) and a right lateralization of the reflexive tasks (pharyngeal). These results are consistent with other studies but are the most precise in terms of temporal course in these conditions. This high precision clearly illustrates MEG’s higher temporal resolution than fMRI, which could only measure a global right insular lateralization [64].

### 4.2. Adaptive Physiology

Besides simple saliva and water studies, some teams studied the adaptive physiology of swallowing through other modifications of the bolus types and volumes, or other factors.

Dziewas’ team is the only one that studied various swallowing condition modifications and their effects on the MEG signal with a consistent self-paced water swallowing protocol over all their studies. They studied the effect of age on cortical activity and showed that elders had a stronger and broader bilateral pericentral activity, predominantly in the β band. In comparison to their usual task, they showed an increased pericentral activity in faster and more challenging swallowing tasks, with specific activation of PMC and parieto-occipital cortex during their hardest task. They illustrated that sensorimotor cortical activity of swallowing is reduced by pharyngeal anesthesia [29] and increased by cold tactile thermal oral stimulation [31]. Following the latter, they tested the sensory effect of capsaicinoids. It appeared that capsaicinoids would have a specific peripheral effect on muscular contraction without specific modification of swallowing cortical activity [43].

Some fNIRS studies also focused on adaptive physiology. In this regard, Kober et al. retrieved a cortical increase in oxyHb concentrations over the IFG in elders [53], similarly to the aforementioned Dziewas study. Mulheren et al. and Lee et al. studied the effect of taste on swallowing cortical activity. In healthy subjects, Mulheren et al. showed no pericentral or premotor (SMA) early effect (2 to 7 s after swallowing) on oxyHb concentrations but found a remanent effect at 17 to 22 s, with significant effect of a sour taste (in comparison to sweetness or water) [48]. Lee et al. focused on the prefrontal activity in healthy and brain-impaired dysphagic subjects. Although they showed globally increased oxyHb concentrations with flavor and decreased oxyHb with sweetness, they found no modifications in brain impaired dysphagic patients [52]. As previously noted, Matsuo’s study with cucumber slices shows the feasibility of fNIRS studies on the ecological meal process, from the plate to the swallowing [54]. It should be noted that they focused only on premotor areas (PMC and SMA) and on cranial sensorimotor cortex, which is likely to exclude the laryngeal cortex and to include superior limb motor cortex. Nevertheless, this is the first neurofunctional study of the whole process of solid swallowing without pause, with other studies being limited to the swallowing part.

Some EEG studies tried to explore the whole process with liquids but introduced pauses between the self-instillation from a glass and the swallowing of the bolus. Cuellar et al.’s cued task comprised a cued self-instillation from cup into the mouth, a 7 s pause then a cued swallowing to reduce arm movement artifacts [19].

The studies of Jestrović et al. with EEG are of note, as they are original and illustrate the microarchitectural characteristics (also called “small-worldness”) of the different global swallowing networks. They used a self-paced multiple successive swallowing from cup without pause between arm and mouth paradigm (but a 2–3 s pause between each swallowing) which made those tasks “self-feeding like”, similarly to Matsuo, in order to study those networks. “Networks” here is plural as they show arguments of different microarchitectural characteristics for different head positions [17,18], bolus types and volumes [16,18,21] and in the presence of distractors [21], suggesting different networks at work. These characteristics could be of use in studying microarchitectural anomalies and better understand pathophysiology of dysphagia and the effect of rehabilitation [17], even though they do not have a “localizationist” value and their future usefulness in clinical conditions needs to be clarified.

To our knowledge, no other neurofunctional imaging studies of swallowing in ecological conditions have been performed with fMRI. The closest to ecological study is the one from Harris et al. with PET [66]. They performed a cued water swallowing task in ecological condition (seated position) during the 18F-FDG uptake and used the remanent fixation to neural network of FDG to perform the PET in lying position. Interestingly, they showed increased glucose metabolism in the left sensorimotor cortex, cerebellum, thalamus, precuneus, anterior insula, left and right lateral postcentral gyrus, and left and right occipital cortex, and decreased glucose metabolism were also seen in the right PMC, right and left sensory and motor association cortices, left posterior insula and left cerebellum. These results are quite consistent with our review, but it must be reiterated that PET being radiative, has been excluded of our review.

### 4.3. Patho-Physiological Contexts

#### 4.3.1. Pathological Descriptions

As previously emphasized, these techniques have been used in pathological conditions. Dziewas’ team is again the most advanced, as they observed that multiple pathological conditions (Amyotrophic lateral sclerosis [36], strokes [35], Parkinson’s disease (PD) [37], etc…) seem to broaden the activity areas, which tends to be more right-sided in those patients. In these pathologic conditions, MEG found activity in the parietal cortex (BA7, 40, 43) but also in premotor (BA6) and prefrontal cortex (BA44, 45, 47). This broadening seems more associated with the presence of dysphagia. Similar prefrontal activity could also be recorded with fNIRS [45] but might be highly dependent on the type and localization of the lesions and obviously on the position of the limited number of optodes of this technique. This general pattern actually shows specificities with each disease. For example, Teismann et al. studied the cortical modifications during swallowing found in cases of stroke. In the case of hemispheric stroke, regardless of dysphagia, they found a higher activity of the Dorsolateral PFC (DLPFC) and insula compared to healthy controls. The presence of dysphagia modifies pericentral activity as hemispheric stroke patients with dysphagia showed a reduction in ipsilateral pericentral activity with no contralateral activity, whereas non dysphagic subjects showed bilateral activity similar to controls. Lastly, brainstem stroke patients displayed a right lateralization of their pericentral activity. With fNIRS, Kober et al. compared the inferior frontal cortex (DLPFC) cortical activity of 2 hemispheric stroke patients, 2 brainstem stroke patients and 2 healthy subjects [45]. They found different results with the 4 patients. The hemodynamic responses (HDR) appeared to be of lower amplitude in cerebral stroke patients and of higher amplitude in brainstem stroke patients, compared to healthy subjects. The authors interpret these as a sign of cortical plasticity after stroke for the swallowing function. Liu’s meta-analysis on fMRI in stroke dysphagic patients during swallowing found slightly different results. Indeed, in patients, they showed an hyperactivation of the left cingulate and precentral gyri and right posterior cingulate gyrus with hypoactivation of the right cuneus and left MFG. This discrepancy could reflect the difference of tasks position (lying down with fMRI and seated with MEG and fNIRS), difference of populations as strokes are different according to the region and their extension, or difference of temporal resolution between MEG (analysis over 200 ms intervals) and fMRI and fNIRS (analysis over a few seconds period).

The only pathology study with EEG compared children with anterior open bite with children with normal denture and found no difference of swallowing EEG activity between the two conditions [22].

#### 4.3.2. Pathophysiological Experiments

These techniques have already been used for pathophysiological studies and pre-therapeutical objectives. These results were not specifically the objective of our review but are still interesting to point out. Yuan et al. used transcranial Direct Current Stimulation (tDCS) and compared EEG results before and after tDCS and showed an increase in swallowing cortical excitability in healthy and cerebral stroke subject with swallowing apraxia. Dziewas’s team also showed an increase in cortical activity in healthy subjects on MEG with tDCS, associated with an improvement of swallowing skills during challenging tasks (e.g., fast swallowing). Based on that, they successfully used tDCS on dysphagic patients through a clinical prospective double-blind protocol, showing its efficiency on cortical activity [41]. They showed similar results in healthy subjects after thermal tactile oral stimulation (used for rehabilitation) [31] and a reduction in cortical activity after pharyngeal electrical stimulation (PES) [40]. They lastly showed that PES could increase cortical activity after pharyngeal anesthesia, which was not possible with tDCS [46].

EEG and fNIRS might also be powerful swallowing rehabilitation tools, through motor imagery and neurofeedback [46,53,67]. Kober et al. are the more advanced with regards to fNIRS neurofeedback of dysphagia. Their preliminary results suggest an ability of fNIRS to stimulate cortical plasticity of the swallowing network, which could lead to dysphagia improvement [46,53]. This actually needs further thorough evaluations and validations, but it is nevertheless a promising treatment.

### 4.4. Limitations and Perspectives

The three techniques showed encouraging results for the exploration of the swallowing function in near-ecologic conditions. However, they both have limitations for use during a mealtime that still needs to be taken into account and overcome.

The easiest tool seems to be fNIRS but it has some limitations. The actual limited number of optodes (the fNIRS captors) implies that one must know the region of interest before the experiment. As most of the activity is located in the caudal pericentral cortex and posterior inferior frontal cortex, it leaves a few optodes to explore other areas. Increasing the number of optodes in future systems should improve this drawback. Moreover, the inaccessibility of the insula and the cingulum limits the use of fNIRS. Some studies used high definition fNIRS systems to explore deeper brain layers (resembling a tomographic technique) but their transferability to swallowing exploration is not certain [68]. One last drawback of about half of the studies is the lack of proof of real HDR measurements. Indeed, almost half of studies only used the oxyHb concentration evolution. It is well known that, with fNIRS, HDR is defined as an inverse variation of oxyHb and deoxyHb [69]. On the contrary, if oxyHb and deoxyHb vary in the same way, it is considered to be non-hemodynamic signal and might be due to movement artifacts. Thus, it is hard to tell if those studies recorded HDR signal or artifacts. This problem also questions the validity of fMRI studies as BOLD (Blood oxygen level dependent) signal also only focus on oxyHb. The measures of both oxyHb and deoxyHb with fNIRS might help to validate or correct previous fMRI results on this matter. Moreover, the major interest of this technique is the low number of trials needed (~10) to record good signal, similarly to fMRI. Its use for motor execution seems promising but still needs more robust studies and its potential for swallowing preparation still needs to be assessed.

For deeper brain studies with more precision in both localization and time course of activity, both EEG and MEG are the best mastered techniques for ecological conditions in both preparation and execution of swallowing. However, the classical need for a lot of trials (50 to 100) is less adapted to classical meal modalities. It should be highlighted that decades of optimizations have allowed for their use with fewer trials (down to 5 [21,67]) but their clinical value is more limited. Another often raised drawback is the important effect of muscular contraction artifacts, but as our review suggests, many analyses can suppress those artifacts within these two techniques.

One way to overcome these limitations could be the use of both EEG or MEG (neuronal signal) and fNIRS (hemodynamic signal) to correlate both results. The EEG/fNIRS association has been used in other fields with promising results [70], but still needs to be evaluated for swallowing purposes. This could improve the data quality and reduce the number of needed trials. The association of the hemodynamic data from fNIRS and neuronal activity from EEG is particularly interesting, as they can be both portable (which is not the case for MEG) and some systems already integrate both signals. The fNIRS itself would also benefit from the short channel technology [71,72] to reduce the effects of non-hemodynamic blood flows, however, that has not been used for swallowing up to now.

Another limitation is the comparison with the gold standard which is the MRI. As we already discussed, when we compare these techniques’ results with those of fMRI, there are some discrepancies that might be due to the position itself. Vertical MRI (called “weight-bearing MRI”) have been used for two decades to study the effect of one’s weight on one’s joints [70]. Their use for functional imaging would allow researchers to differentiate the proper effect of the lying down position but the technology has not been used in this way up to now as the scanner’s power is limited to 1.5T [73].

## 5. Conclusions

Neurofunctional imaging of the swallowing function in ecologic condition is possible through EEG, MEG and fNIRS but still needs to be improved. As each technique has its benefits and drawbacks, the improvement could well arise from multi-signal explorations to allow for meal-time analysis in the future. This will help to improve physiology and pathology comprehension and might lead to the rehabilitation of dysphagia in these subjects.

## Figures and Tables

**Figure 1 jcm-11-05480-f001:**
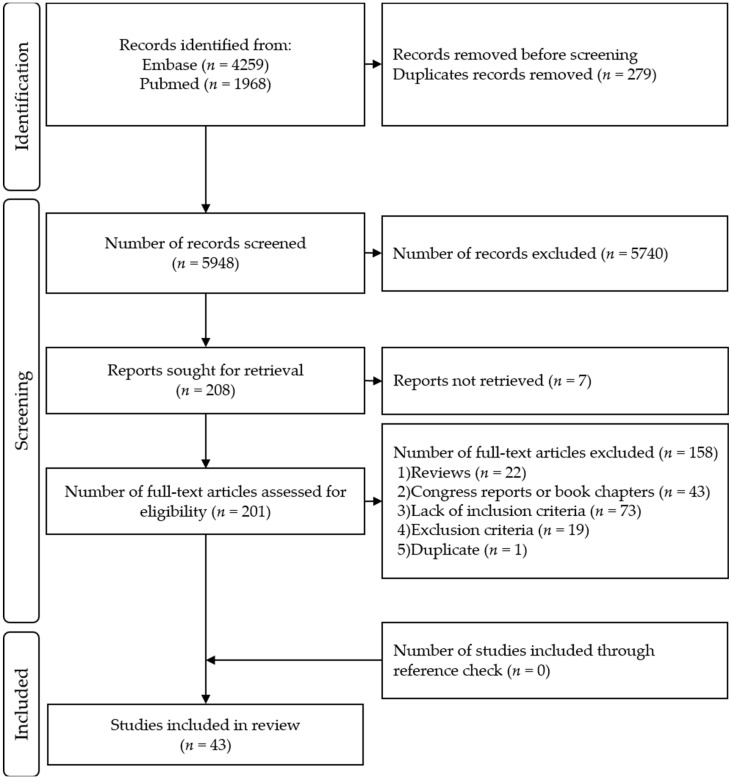
PRISMA Flow Diagram.

**Figure 2 jcm-11-05480-f002:**
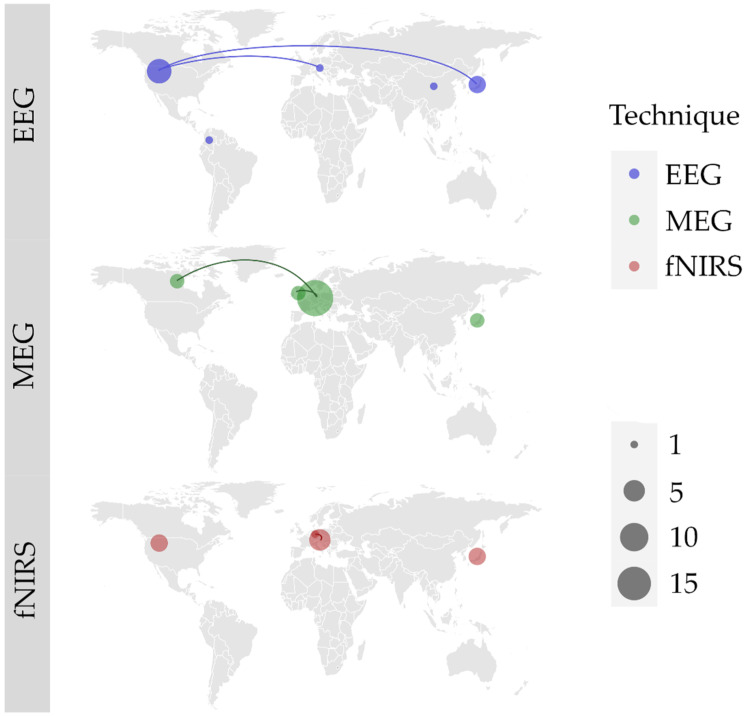
World locations of author’s affiliations, and collaborations (lines) between research teams for each of the three identified techniques: electroencephalography (EEG); magnetoencephalography (MEG); functional near infrared spectroscopy (fNIRS). Size of the point represents the number of articles published by each and every country.

**Figure 3 jcm-11-05480-f003:**
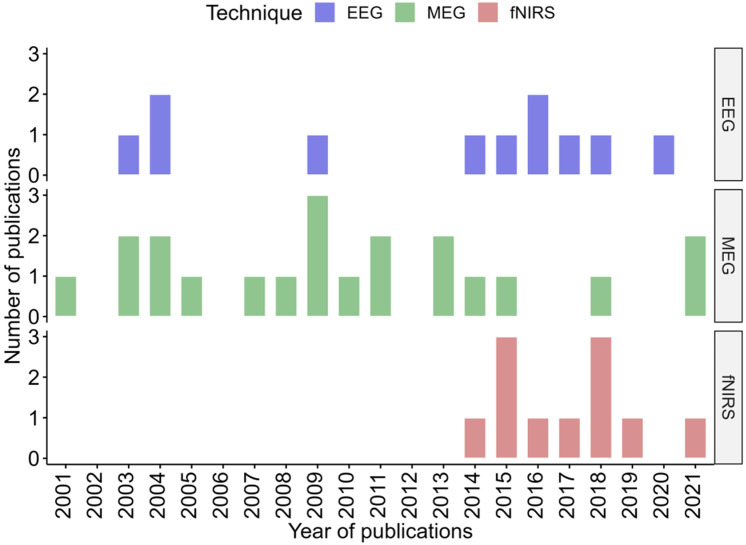
Distribution timeline, in years, of all included publication for each technique.

**Table 1 jcm-11-05480-t001:** Search strategies.

Database	Search Terms (Subject Headings and Free Text Words).	Number of Records
**Embase**	(Dysphagia/OR Swallowing/) AND (functional near-infrared spectroscopy/OR functional neuroimaging/OR spectrophotometry/OR spectroscopy/OR electroencephalogram/OR magnetoencephalography/OR automated pattern recognition/OR brain computer interface/OR brain blood flow/OR brain electrophysiology/OR brain mapping/OR brain metabolism/OR brain cortex/OR brain/OR computer assisted diagnosis/OR hemoglobin/OR deoxyhemoglobin/OR oxyhemoglobin/OR brain radiography/OR electroencephalography/OR hemodynamics/OR oxyhemoglobin/OR neurovascular coupling/OR brain computer interface/OR noninvasive brain-computer interface/OR fluorescence imaging/OR oxygen/)	4259
**Pubmed**	(“Deglutition”[Mesh] OR “Deglutition Disorders”[Mesh]) AND (“Functional Neuroimaging”[Mesh] OR “Spectroscopy, Near-Infrared”[Mesh] OR “Spectroscopy, Fourier Transform Infrared”[Mesh] OR “Proton Magnetic Resonance Spectroscopy”[Mesh] OR “Carbon-13 Magnetic Resonance Spectroscopy”[Mesh] OR “Dielectric Spectroscopy”[Mesh] OR “Photoelectron Spectroscopy”[Mesh] OR “Terahertz Spectroscopy”[Mesh] OR “Spectroscopy, Electron Energy-Loss”[Mesh] OR “Magnetic Resonance Spectroscopy”[Mesh] OR “Electron Spin Resonance Spectroscopy”[Mesh] OR “Spectrometry, Mass, Secondary Ion”[Mesh] OR “Single Molecule Imaging”[Mesh] OR “Nuclear Magnetic Resonance, Biomolecular”[Mesh] OR “Spectrometry, Mass, Matrix-Assisted Laser Desorption-Ionization”[Mesh] OR “Spectrometry, Mass, Fast Atom Bombardment”[Mesh] OR “Spectrum Analysis, Raman”[Mesh] OR “Mass Spectrometry”[Mesh] OR “Spectrometry, Fluorescence”[Mesh] OR “Spectrophotometry, Atomic”[Mesh] OR “Ultraviolet Rays”[Mesh] OR “Infrared Rays”[Mesh] OR “Terahertz Radiation”[Mesh] OR “Spectrophotometry”[Mesh] OR “Spectrophotometry, Ultraviolet”[Mesh] OR “Spectrophotometry, Infrared”[Mesh] OR “Spectrophotometry, Atomic”[Mesh] OR “Spectrometry, Fluorescence”[Mesh] OR “Electroencephalography”[Mesh] OR “Electroencephalography Phase Synchronization”[Mesh] OR “Electrocorticography”[Mesh] OR “Magnetoencephalography”[Mesh] OR “Pattern Recognition, Automated” [Mesh] OR “Brain-computer interfaces” [Mesh] OR “Brain mapping” [Mesh] OR “Brain Diseases, Metabolic, Inborn”[Mesh] OR “Cerebral Cortex”[Mesh] OR “Brain”[Mesh] OR “Brain/blood”[Mesh] OR “Brain/blood supply”[Mesh] OR “Brain/diagnostic imaging”[Mesh] OR “Brain/metabolism”[Mesh] OR “Diagnosis, Computer-Assisted” [Mesh] OR “Hemoglobins”[Mesh] OR “deoxyhemoglobin” [Supplementary Concept] OR “Oxyhemoglobins”[Mesh] OR “Electroencephalography Phase Synchronization”[Mesh] OR “Electrocorticography”[Mesh] OR “Hemodynamics”[Mesh] OR “Oxyhemoglobins”[Mesh] OR “Neurovascular Coupling” [Mesh] OR “Brain-Computer Interfaces”[Mesh] OR “Oxygen”[Mesh])	1968

**Table 2 jcm-11-05480-t002:** Methodological quality rating according to the QualSyst critical appraisal tool [11] and the NHMRC level of evidence [10] of the 43 included articles.

Technique	Reference	QualSyst (%) *	Methodology Quality *	NHMRC Level of Evidence **
EEG	M. L. Huckabee 2003 [12]	17/20 (85%)	Strong	IV
	T. Satow 2004 [13]	18/20 (90%)	Strong	IV
	K. Hiraoka 2004 [14]	15/22 (68%)	Good	IV
	T. Nonaka 2009 [15]	19/22 (86%)	Strong	IV
	I. Jestrović 2014 [16]	20/24 (83%)	Strong	IV
	I. Jestrović 2015 [17]	19/22 (86%)	Strong	IV
	I. Jestrović 2016 [18]	20/22 (91%)	Strong	IV
	M. Cuellar 2016 [19]	18/22 (82%)	Strong	IV
	Y. Yuan 2017 [20]	22/26 (85%)	Strong	III2
	I. Jestrović 2018 [21]	19/22 (86%)	Strong	IV
	C. Restrepo 2020 [22]	18/24 (75%)	Good	III2
MEG	R. Loose 2001 [23]	16/22 (73%)	Good	IV
	S. Abe 2003 [24]	14/22 (64%)	Good	IV
	R. Dziewas 2003 [25]	18/22 (82%)	Strong	IV
	P. L. Furlong 2004 [26]	18/22 (82%)	Strong	IV
	Y. Watanabe 2004 [27]	16/20 (80%)	Strong	IV
	R. Dziewas 2005 [28]	19/22 (86%)	Strong	IV
	I. K. Teismann 2007 [29]	20/24 (83%)	Strong	IV
	I. K. Teismann 2008 [30]	19/22 (86%)	Strong	III3
	I. K. Teismann 2009 [31]	21/24 (88%)	Strong	IV
	I. K. Teismann 2009 [32]	20/22 (91%)	Strong	IV
	R. Dziewas 2009 [33]	20/22 (91%)	Strong	III3
	I. K. Teismann 2010 [34]	20/22 (91%)	Strong	III3
	I. K. Teismann 2011 [35]	21/22 (95%)	Strong	III3
	I. K. Teismann 2011 [36]	21/22 (95%)	Strong	III3
	S. Suntrup 2013 [37]	21/22 (95%)	Strong	III3
	S. Suntrup 2013 [38]	27/28 (96%)	Strong	III1
	S. Suntrup 2014 [39]	21/22 (95%)	Strong	III3
	S. Suntrup 2015 [40]	26/28 (93%)	Strong	III1
	S. Suntrup-Krueger 2018 [41]	24/26 (92%)	Strong	II
	P. Muhle 2021 [42]	24/28 (86%)	Strong	IV
	S. Suntrup-Krueger 2021 [43]	20/22 (91%)	Strong	IV
fNIRS	S. E. Kober 2014 [44]	20/22 (91%)	Strong	IV
	S. E. Kober 2015 [45]	17/22 (77%)	Good	III3
	S. E. Kober 2015 [46]	23/28 (82%)	Strong	IV
	K. Inamoto 2015 [47]	17/22 (77%)	Good	IV
	R. Mulheren 2016 [48]	22/22 (100%)	Strong	IV
	R. Mulheren 2017 [49]	22/24 (92%)	Strong	III2
	E. Kamarunas 2018 [50]	23/24 (96%)	Strong	IV
	S. E. Kober 2018 [51]	21/22 (95%)	Strong	IV
	J. Lee 2018 [52]	20/22 (91%)	Strong	III3
	S. E. Kober 2019 [53]	24/28 (86%)	Strong	III2
	M. Matsuo 2021 [54]	17/22 (77%)	Good	IV

* Methodological quality: strong (>80%); good (60–79%); adequate (50–59%); and poor (<50%). ** NHMRC hierarchy: Level 1, systematic reviews; level II, randomized control trials; level III-1, pseudo-randomized control trials; level III-2, comparative studies with concurrent controls and allocation not randomized (cohort studies), case control studies, or interrupted time series with a control group; level III-3, comparative studies with historical control, two or more single-arm studies, or interrupted time series without a control group; level IV, case series.

**Table 3 jcm-11-05480-t003:** Technical characteristics of EEG, fNIRS, MEG: methodology, swallowing sequences and measure of interest as reported in the included studies, compared to fMRI. See Appendix A for the complete results.

Technique and Signal Type	Spatial and Temporal Resolutions	Studied Phase of Swallowing Act	Type of Analyses ^a^ (Number of Studies)	Median Time Window from Swallowing Act Onset (ms) [Start:End]	Task Iterations (Min-Max)	Regions of Interest ^c^
**EEG** Whole head cerebral electrical activity	1 mm 200–500 Hz	Preparation	Topographic (n = 3) Topographic (n = 1)	−5000: +1000 −1024: 0	50–480 20	Cz, Fz, FCz, Pz, P3, P4, C3, C4, F4, T5, T5
		Preparation and execution	Topographic (n = 1) Network micro-architecture (n = 4) Topographic (n = 1)	−1500: +1000	50	C3, C4, Cz

				-	5	Whole head


				-	1	C3, C4
		Execution	Topographic (n = 1)	−1000: +3000	80	C3, C4,Cz
**fNIRS** Targetted optical hemoglobins concentrations HDR	2–3cm 7–50Hz (HDR > 1 s)	Execution	Topographic (n = 6) Topographic (n = 5)	−5000: + 37,500 −5000: +25,000	3–30 10–20	Caudal pericentral Cx PMC, SMA, PFC Inferior frontal gyrus
**MEG** Whole head cerebral electro-magnetic activity	1 mm 400–600 Hz	Preparation	Tomographic (n = 2)	−2500: +500	30–50	**Whole head** Cingulate gyrus SMA Insula Inferior frontal gyrus
		Preparation and execution	Tomographic (n = 16)	−3000: + 2000	40–100	**Whole head** Pericentral Cx PMC, SMA, PFC Parietal Cx Insula
		Execution	Topographic (n = 1) Tomographic (n = 2)	−500: +1500 0	100 20–100	**Whole head**: No result **Whole head** Pericentral Cx Parietal Cx
**fMRI** ^b^ Whole head BOLD signal HDR	3–5 mm 14.5 Hz (HDR > 1 s)	Preparation and execution	Tomographic	-	10	Primary sensorimotor Cx, PMC, SMA, PFC, Heschl’s gyrus, cingulate gurus, insula, Broca’s areas, superior temporal gyrus, precuneus

BOLD: Blood oxygen level dependent; Cx: Cortex; HDR: Hemodynamic response; PFC: Prefrontal cortex; PMC: Premotor cortex; SMA: Supplementary motor area; ^a^ A topographic analysis is limited to the superficial cortical layer, whereas a tomographic method can explore deeper structures (e.g., grey nuclei, insula, cingulum). ^b^ fMRI data as reported in reviews by Malandraki et al. [55] and Ludlow et al. [1,4]. ^c^ The region of interest is defined based on the technique data reported in the included studies. Only the most relevant electrode/channels/areas across studies were reported. For MEG studies, the whole head coverage was reported as necessary to calculate the source of the activity. For details and equivalence between the 10-10 system, Brodmann areas and gyrus, see Appendix A.

**Table 4 jcm-11-05480-t004:** EEG studies data. See complete version in Appendix A.

EEG Reference	Objectives	Tasks and/or Conditions (Bold: Eligible for the Review)	Review Conclusions	Locus of Interest 10-10 System ^a^
**Physiology**				
Huckabee 2003 [12]	**To evaluate the role of the cerebral cortex in the motor planning and initiation of deglutitive behavior, focusing on Bereischaftpotential (BP)** **To investigate whether the act of swallowing utilizes cortical motor planning under the condition of volitional swallowing.**	**Task 1: Self-paced breathing 5 s pause-before volitional saliva swallowing “with effort”** Task 2: Finger press movement	Swallowing evokes a 1 phase BP that can be measured on Cz FCz FC1z FC2z during the [−5000 ms:0 ms] time window Trend for lower amplitude for swallowing at 4 time points (*p* < 0.10) BP for finger tapping was not significantly earlier than for swallowing No lateralization	Cz FCz FC1z FC2z
Satow 2004 [13]	**To clarify whether the hemispheric dominance can be determined in the preparatory period of swallowing or not.**	**Task 1: Self-paced 2–3 mL water swallowing** Task 2: Tongue protrusion	Earlier BP with swallowing (*p* = 0.012), maximum at vertex midline (Cz) during the [−3000 ms:0 ms] time window. No lateralization.	Cz
Hiraoka 2004 [14]	**To differentiate among the cortical activities of motor preparation, execution, and regulation of swallowing using Movement Related Cortical potentials (MRCPs)** **To document the MRCPs associated with saliva and water swallowing**	**Task 1: Self-paced volitional saliva swallowing** **Task 2: Self-paced volitional water swallowing from glass in right hand (with 10s rest between infusion and swallowing)**	MRCP/BP amplitude is greater with saliva than water (*p* = 0.035) and can be measured on C3, C4 and CZ within a [−1500 ms:0 ms] time window. Positive potential amplitude during execution is greater with water than saliva (*p* = 0.048) and can be measured on C3, C4 and CZ within a [0 ms:1000 ms] time window. No lateralization.	C3 C4 Cz
Nonaka 2009 [15]	**To compare the waveforms of contingent negative variation (CNV) associated with the command swallowing task with those of movement related cortical potential (MRCP) associated with the volitional (self-paced) swallowing task in healthy adults.** To elucidate the effects of human swallowing training on brain activities preceding the onset of swallowing.	**Task 1: Self-paced breathing** **4–6 s pause-preceded volitional saliva swallowing “with effort”** **Task 2: Auditory cued Breathing 4 s pause-preceded saliva swallowing task**	Negative preparatory potentials (CNV and MRCP) can be measured on Fz, Cz, Pz, C3, and C4 (mostly Cz and Fz) up to 2 s before the swallowing muscular movement. Their onset time depends on the task type (cued or volitional). CNV amplitude stronger than MRCP amplitude (*p* < 0.01) Stronger CNV’ at Cz (*p* < 0.05)	Fz Cz Pz C3 C4
Cuellar 2016 [19]	**To use Independent Component Analysis (ICA) to identify bilateral sensorimotor mu components and infrahyoid muscle components in the primarily reflexive pharyngeal and esophageal phases of swallowing** and a voluntary tongue-tapping task **To use event-related spectral perturbation (ERSP) to provide measures of sensorimotor activity across time that can be referenced to infrahyoid muscle activity.** To validate further use of this non-invasive means of measuring neural responses.	**Task 1: Visually cued self-administered 5 mL water swallowing** Task 2: Tongue tapping	Swallowing execution evokes bilateral Mu ERD rythm localized in BA4 and BA6 with right lateralization and can be measured in a [0 ms:2000 ms] time window in α and β bands	C3 C4 Cz
**Adaptive physiology**				
Jestrović 2014 [16]	**To investigate the stationarity of the EEG signal during swallowing and the effect of sex, age, different brain regions, and the viscosity of the swallowed liquids.**	**Task 1: Self-paced Saliva swallowing** **Task 2: Self-paced water (1 cp) swallowing from cup** **Task 3: Self-paced honey (150 cP) swallowing from cup** **Task 4: Self-paced nectar (400 cP) swallowing from cup**	Whole head EEG shows that swallowing signal is non-stationary and needs specific methods to be studied. INS increase with viscosity and is the highest with saliva (*p* < 0.01). Male participants exhibited higher non-stationarity (*p* < 0.01), except for water swallows.	na
Jestrović 2015 [17]	**To compare the small-world properties of brain networks for swallowing in two head positions: the neutral or natural position, and the chin-tuck head position,**	**Task 1: Self-paced saliva swallowing neutral position** **Task 2: Self-paced saliva swallowing chin-tuck head position**	Neutral and chin tuck position swallowing networks display small-world characteristics and seems to differ for some features (e.g., clustering coefficient and characteristic path length). Differences are found in α (inhibitory cognitive and motor tasks) and γ bands (performance of cognitive and motor tasks)	na
**Jestrović** **2016** [18]	**To compare the brain networks in term of small world properties, according to swallowing of various fluid viscosities, as well as between swallowing in the neutral and chin-tuck head positions**	**Task 1: Self-paced water (1 cP) swallowing** **Task 2: Self-paced nectar (150 cP) swallowing** **Task 3: Self-paced honey (400 cP) swallowing** **Task position A: Self-paced neutral position swallowing (with either aforementioned thickness)** **Task position B: Self-paced chin-tuck position swallowing (with either aforementioned thickness)** Every task (1, 2,3) was performed in both positions (A and B)	Significant differences in the brain networks in terms of clustering coefficient, characteristic path length and small-worldness depending on the bolus thickness (in α, β, γ, δ, θ bands, *p* < 0.05) and the head position (α, β, γ band, *p* < 0.05) The functional brain network activated during swallowing has small-world properties.	na
**Adaptive physiology**				
Jestrović 2018 [21]	**To investigate the effects of external distraction on brain activity during swallowing.**	**Task 1/Condition 1: Self-paced 1 mL water swallow without distractor** **Task 1/Condition 2: Self-paced 1 mL water swallow with distractor** **Task 2/Condition 1: Self-paced 5 mL water swallow without distractor** **Task 2 /Condition 2: Self-paced 5 mL water swallow with distractor** **Task 3/Condition 1: Self-paced 10 mL water swallow without distractor** **Task 3/Condition 2: Self-paced 10 mL water swallow with distractor**	Significant differences in the brain networks in terms of: clustering coefficient, characteristic path length and small-worldness depending on the presence of distractors and the swallowed volume (in α, β, γ, δ, θ bands, *p* < 0.05) The brain network is different for no-distraction swallowing compared with the brain network constructed during swallowing with distraction These results showed differences in the swallowing of boluses of various volumes in all frequency bands of interest.	na
**Patho-Physiology**				
Yuan 2017 [20]	**To investigate the effect of tDCS on swallowing apraxia and cortical activation in stroke patients**	**Task 1: Auditory cued volitional water swallowing pre-tDCS** **Task 2: Transnasally provoked water swallowing pre-tDCS** Task 1 post-tDCS: Auditory cued water volitional swallowing Task 2 post tDCS: Transnasally provoked water swallowing transnasally Task 3: Rest pre-tDCS and post-tDCS	Regardless of tDCS Physiology: Volitional water swallowing increases ApEn n F4, C4, P3, P4, and T5 (*p* < 0.01) Reflexive swallowing increases ApEn in C4 (*p* < 0.01) Pathology: Volitional water swallowing did not modify ApEn Reflexive swallowing increased ApEn in left-sided regions (C3, P3, T5)	P3 P4 C3 C4 F4 T5 T5
**Pathology**				
Restrepo 2020 [22]	**To determine the activity of the brain cortex of children with Anterior Open Bite (AOB) at rest and during phonation and deglutition** To evaluate the association of intelligence quotient (IQ), attention [Test of Variables of Attention (TOVA)], and oxygen saturation with brain activity in subjects with AOB.	**Task 1: AOB group-10s self-paced swallowing from glass of water** **Task 1: Non-AOB group-10s self-paced swallowing from glass of water** Task 2: AOB group-50s phonation task Task 2: Non-AOB group-50s phonation task Task 3: AOB group—Rest Task 3: Non-AOB group-Rest	There was no difference between the two groups for the swallowing execution The only difference was found during the rest task between the two groups on C3 and C4 electrodes with a higher left-sided activity in the AOB group in α/θ band (*p* = 0.05) and on α band (*p* = 0.02).	Rest: C3 C4

AOB: Anterior open bite; BP: Bereitschaftspotential; CNV: Contingent Negative Variation; MRCP: Movement correlated cortical potential, contains Bereischaftspotential and Negative slope; Na: not available; NF: Neurofeedback; NS: non significative; R/L: Right/Left; tDCS: Transcranial Direct-Current Stimulation; Bold text in objectives and tasks reflects the parts included in our review. ^a^ Correspondence between 10-10 system, Brodman areas and anatomical gyri are available in Appendix A. They were performed according to the article data but also Scrivener and Reader 2022 [56], Okamoto 2004 [57] and http://bioimagesuite.com (based on Lacadie et al. 2008 [58], accessed on 22 April 2022) when coordinates were available.

**Table 5 jcm-11-05480-t005:** fNIRS studies data. See complete version in Appendix A.

fNIRS Reference	Objectives	Tasks and/or Conditions (Bold: Eligible for the Review)	Review Conclusions	Locus of Interest Anatomical Gyri ^a^
**Physiology**				
Kober 2014 [44]	**To examine cortical correlates of motor execution and imagery of swallowing using NIRS.**	**Task 1: Self-paced volitional 5–6 water swallowing through oral tube** Task 2: Self-paced motor imagery of 5–6 swallowing through oral tube	Swallowing activity localized in the bilateral inferior frontal gyri, measured within a [0 ms:+25,000 ms] time window (*p* < 0.05)	Inferior frontal gyrus Pars opercularis
Inamoto 2015 [47]	**To examine cerebral blood volume dynamics during volitional swallowing using multi-channel fNIRS** **To identify the specific regions of the cerebral cortex that exhibited activation.**	**Task: Orally cued 5 mL water swallowing**	Using the OxyHb concentration changes, it is possible to visualize the swallowing cortical evoked CBF in the posterior frontal region and its surroundings (*p* < 0.05)	Precentral gyrus Postcentral gyrus Superior temporal gyrus Middle temporal gyrus Left middle frontal gyrus Inferior frontal gyrus Supramarginal gyrus
Kamarunas 2018 [50]	**To determine the timing and amplitude characteristics of cortical activation patterns in** **the right and left precentral motor and postcentral somatosensory regions during spontaneous reflexive saliva swallows using fNIRS.**	**Task: Spontaneous swallowing during rest without instruction**	In the four region, the mean peak times are situated during the [3–4 s] interval (early response) and during the [13–22.5 s] interval (late response). Spontaneous not cued swallowing evokes an early cortical response peak during the [0:8 s] period. Left S1 response was the earliest at onset (−2 s, *p* < 0.008) with stronger responses. This response is non-significantly followed by responses of right M1, right S1 and last left M1. Spontaneous un-cued swallowing evokes a late cortical response [8–35 s]. Time course across the regions was not significant for the late peak. The strongest HbO_2_ change were found in left S1 in comparison to left M1 (*p* < 0.005) regions during the early peak The four regions’ activity seems independent, as activity correlations were insufficient, with the strongest correlations between left S1 and right M1 (*r* = 0.63) and both M1 (*r* = 0.63)	Precentral gyrus (M1) Postcentral gyrus (S1)
**Physiology**				
Kober 2018 [51]	**To investigate whether NIRS is sensitive enough to reveal differences in the hemodynamic response over the bilateral IFG between swallowing saliva and water in healthy adults.** **To compare the hemodynamic response over the two hemispheres**	**Task 1: Self-paced volitional 5 to 6 water swallowing through oral tube** **Task 2: Self-paced volitional 5 to 6 saliva swallowing**	Strongest swallowing evoked response is located bilaterally in the inferior frontal region in pars opercularis (False discovery rate, *p* < 0.10) Differences between water and saliva with higher oxyHb responses for saliva (*p* < 0.05)	Inferior frontal gyrus Pars opercularis
Matsuo 2021 [54]	**To investigate the cerebral hemodynamics associated with the MI and ME of a self-feeding activity with chopsticks**	**Task 1: Stopwatch-cued volitional cucumber eating with chopsticks** Task 2: Assistant orally cued motor imagery of cucumber eating with chopsticks	Swallowing execution evokes a typical oxyHb HDR in SMC, PFC and pre-SMA and a oxyHb decrease in SMA and PMA between [5:25 s] after onset (*p* < 0.05)	PMC Pre-SMA, SMA Sensorimotor Cx PFC
**Adaptive physiology**				
Kober 2015 [46]	To address the question whether both hemodynamic parameters of fNIRS, oxy- and deoxy-Hb, can be modulated voluntarily by means of real-time neuro-feedback (NF), when participants imagine swallowing. **To study the effects of NIRS-based NF training on swallowing related brain activation patterns, measuring the cortical correlates of ME** and MI of swallowing before and after NF training.	**Task 1: Before NF Self-paced volitional 5 to 6 water swallowing through oral tube before NF** **Task 1: After NF Self-paced volitional 5 to 6 water swallowing through oral tube** Task 2: Before NF—Self-paced motor imagery of 5 to 6 swallowing through oral tube Task 2: After NF—Self-paced motor imagery of 5 to 6 swallowing through oral tube	Strongest swallowing evoked response is located bilaterally in the inferior frontal region and measured in a [0 ms:+25,000 ms] time window (*p* < 0.01) This response can be enhanced after deoxyHb Neurofeedback (*p* < 0.05)	Inferior frontal gyrus
Mulheren 2016 [48]	**To determine whether swallowing function and hemodynamic responses differ in response to different tastes (sour and/or sweet) with mediation by genetic taster status.** To study the effect of the presence/absence of a supplemental slow, steady water infusion on both swallowing pace and hemodynamic responses of the primary motor cortex	**Task 1: Self-paced swallowing 3 mL bolus medium sour** **Task 2: Self-paced swallowing 3 mL bolus strong sweet swallowing** **Task 3: Self-paced swallowing 3 mL bolus deionizes water swallowing** **Task 4: Self-paced swallowing 3 mL bolus sour + water infusion (0.08 L/min) swallowing** **Task 5: Self-paced swallowing 3 mL bolus deionized + water infusion (0.08 L/min) swallowing**	Swallowing evoked an early activity peak between [2–7 s] in M1, S1 and SMA that is not influenced by the taste (*p* < 0.05) Swallowing evoked a late activity [17–22 s] influenced by the taste, the highest activity being obtained with sour taste (*p* < 0.05) The oxyHb of the bilateral M1, S1 and SMA were similar during the early peak During the late peak, oxyHb was significantly greater in M1 and in S1, but was similar in SMA and the dummy region (*p* < 0.05)	Precentral gyrus (M1) Postcentral gyrus (S1) SMA
**Adaptive physiology**				
Mulheren 2017 [49]	**To study the effects of different cervical vibrations protocols** (different frequencies, either continuous or pulsed) on: -the fundamental frequency of the voice during stimulation in comparison with voicing without stimulation. -the regulation of brain stem control of swallowing through the swallowing frequency. **-the cortical swallowing network on fNIRS recordings during stimulation epochs** **To compare the cortical effects of vibratory stimulation during stimulation or between stimulation periods during 20-min stimulation conditions in comparison with sham conditions.**	**Condition 1: Spontaneous Swallowing during 10 s cervical vibratory stimulation (8 different frequency conditions) 0–30 s without instruction** **Condition 2: Spontaneous Swallowing after 10 s cervical vibratory stimulation (8 different frequency conditions) 30–45 s without instruction** **Condition 3: Spontaneous Swallowing during sham stimulation 0–30 s without instruction** Condition 4: Control cortical activity during 10 s vibration (regardless of swallowing)	Early HDR [4:7 s] detected in both M1 and S1 and late activity [14:17 s] Activity increased in both early and late response with vibrations compared to sham, with varying lateralization *p* < 0.05	Precentral gyrus (M1) Postcentral gyrus (S1)
Kober 2019 [53]	To compare the trainability of hemodynamic parameters between healthy young and older individuals within one neurofeedback training session. **To investigate if NIRS signal change during executing and imagining swallowing movements is comparable between young and older individuals when no real-time feedback of brain signals is provided**.	**Task 1: Young group—Self-paced volitional 5 to 6 saliva swallowing before NF** **Task 1: Older group—Self-paced volitional 5 to 6 saliva swallowing before NF** Task 2: Young group—Self-paced motor imagery of 5 to 6 saliva swallowing before NF Task 2: Older group—Self-paced motor imagery of 5 to 6 saliva swallowing before NF	During the swallowing task oxyHb was significantly greater on the left IFG than on the right IFG in the 2 groups (*p* < 0.05). No significant difference between young and older subjects (slightly stronger response in younger subjects)	Inferior frontal gyrus Pars opercularis
**Patho-physiology**				
Lee 2018 [52]	**To investigate prefrontal cortex activity using NIRS, in healthy volunteers and dysphagia patients during swallowing of sweetened/unsweetened and flavored/unflavored jelly** **To determine if taste and flavor stimuli modulate prefrontal cortex function in dysphagia patients.**	**Task 1: Self-paced swallowing of unflavored/unsweetened 2 mL jelly by straw** **Task 2: Self-paced swallowing of unflavored/sweetened 2 mL jelly by straw** **Task 3: Self-paced swallowing of flavored/unsweetened 2 mL jelly by straw** **Task 4: Self-paced swallowing of flavored/sweetened 2 mL jelly by straw**	In healthy subjects -An early prefrontal oxyHb response to swallowing is measured at about 10 s -A late peak is seen at about 26 s -Sweetness decreases the responses (*p* < 0.001); flavor increases the response (*p* < 0.001). No peak in dysphagic subjects. Comparing both groups’ responses, unsweetened jelly evoked higher responses in controls (*p* < 0.01)	Prefrontal Cx (Superior frontal gyrus, Medial frontal gyrus)
**Pathology**				
Kober 2015 [45]	**To use NIRS to examine the cortical correlates of swallowing** **in patients with dysphagia.** **To compare the brain activation patterns associated with saliva swallowing between dysphagia patients and healthy-matched controls, in terms of time course and topographical distribution of the hemodynamic signal change (oxy-Hb and deoxy-Hb) during swallowing**. To determine the extent to which Motor imagery (MI) and Motor Execution (ME) of swallowing lead to comparable brain activation patterns in stroke patients.	**Task 1: Self-paced volitional saliva swallowing 3 times—Controls** **Task 1: Self-paced volitional saliva swallowing 3 times—Cerebral stroke patients** **Task 1: Self-paced volitional saliva swallowing 3 times—Brainstem stroke patients** Task 2: Self-paced motor imagery of saliva swallowing 3 times—Controls Task 2: Self-paced motor imagery of saliva swallowing 3 times—Cerebral stroke patients Task 2: Self-paced motor imagery of saliva swallowing 3 times—Brainstem stroke patients	The strongest swallowing activity is localized in the bilateral inferior frontal gyri (*p* < 0.1), measured within a [0 ms:+20,000 ms] time window with peak at 15 s Cerebral stroke patients show less activation than controls with later peak (*p* < 0.1) Brainstem stroke patients show stronger activation than controls with larger region of activity (*p* < 0.1)	Inferior frontal gyrus Pars opercularis

Cx: Cortex; Dysph.: Dysphagic; M1: Primary motor cortex; NF: Neurofeedback; NS: non significative; PFC: Prefrontal cortex; PMC: Premotor cortex; R/L: Right/Left; SMA: Supplementary motor area; S1: Primary sensory motor cortex; ^a^ Correspondence between 10-10 system, Brodman areas and anatomical gyri are available in Appendix A. They were performed according to the article data but also Scrivener and Reader 2022 [56], Okamoto 2004 [57] and http://bioimagesuite.com (based on Lacadie et al. 2008 [58], accessed on 22 April 2022) when coordinates were available.

**Table 6 jcm-11-05480-t006:** MEG studies data. See complete version in Appendix A.

MEG Reference	Objectives	Tasks and/or Conditions (Bold: Eligible for the Review)	Review Conclusions	Locus of Interest Brodmann Areas ^a^
**Physiology**				
Loose 2001 [23]	**To study the sources of activation evoked by active tongue movement employing MEG** **To identify the locality of the major contributors**	**Task 1: Self-paced 5 mL water swallowing** Task 2: Tongue protraction	No cortical source found	None
Abe 2003 [24]	**To investigate whether the decision to drink is made just before swallowing, using a 306-channel whole-head neuromagnetometer.**	**Task: Self-paced 1 mL water bolus by tube**	The magnetic dipole during the swallowing preparation is bilaterally located in the cingulate gyrus and SMA and is active between [−1500 and −1100 ms] before the volitional swallowing muscular activation.	BA24,32 BA6
Dziewas 2003 [25]	**To study cortical activation during volitional and reflexive water swallowing with whole-head MEG and synthetic aperture magnetometry** **(SAM)** **To compare the cortical representation of swallowing with that of a swallow-related but less complex movement task with an added tongue movement paradigm was included in the study design.**	**Task 1: Self-paced volitional 10 mL/min water swallowing by oral tube** **Task 2: Provoked 10 mL/min water swallowing by transnasal tube with volitional tongue movement** Task 3: Tongue propulsion Task 4: Resting stage	The most prominent and consistent activity (α and β ERD) is located in bilateral BA 1,2,3,4,7 in volitional preparation and execution (except only left-sided BA7 activity) Insula and frontal operculum activity (θ, low γ and high γ ERS) is specifically linked to preparation and execution of swallowing in this experiment Pharyngeal reflexive θ band ERS responses are located on left-sided over BA9 for preparation and BA7 for execution	BA1,2,3 BA4 BA7 BA9 BA13 BA44
Watanabe 2004 [27]	**To investigate serial positional changes in the entire activity areas in the cerebral cortex with time until the initiation of swallowing** **movement.** **To define the spatiotemporal relations among regions of the brain involved in the central initiation of human voluntary swallowing using the MEG technique with a larger subject size.**	**Task 1: Assistant administered cued 3 mL water by tube** Task 2: Right middle finger extension	Swallowing preparation evoked bilateral responses located in ACC, PCC, MFG, IFG and Insula. The mesured sequence is PCC > SMA > ACC > SFG > MFG > IFG > Insula but only PCC and Insula had significant different onset times (*p* < 0.003) Insula and IFG activity where more consistent for swallowing than for finger extension The swallowing activity is measurable during the [−2375 ms:−1055 ms] time window	BA31, 23 BA13 BA44 BA 25, 24, 32, 33
Furlong 2004 [26]	**To use MEG to dissociate the relative cortical contributions of each of the separable components of swallowing in the sensorimotor sequence** **To identify the spatio-temporal characteristics of cortical activation during swallowing.** **To enhance our appreciation of the relevance of cortical regions to swallowing and provide insight into the mechanisms underlying dysphagia after cerebral injury.**	Task 1: Assistant administered 5 mL water in mouth by tube (no swallowing) **Task 2: Cued 5 mL water swallowing** Task 3: Cued tongue pressure Task 4: Resting state	Swallowing execution evokes an activation pattern shifting from caudal pericentral cortex activation to superior postcentral gyri and paracentral lobule In this study, activation (ERD) during swallowing appears more right-sided	BA 3, 1, 2 BA4 BA5 BA40
**Physiology**				
Dziewas 2005 [28]	**To apply whole-head MEG in order to study the cortical processing of esophageal sensation in healthy humans**	**Task 1: Self-paced volitional 10 mL/min water swallowing by oral tube** Task 2: Direct esophageal stimulation	During volitional swallowing, β and α activity is left lateralized within the primary sensorimotor cortex.	BA1,2,3 BA4
Teismann 2009 [32]	**To investigate the temporal characteristics of human swallowing in healthy subjects by means of whole-head MEG and SAM.**	**Task: Self-paced volitional 10 mL/min water swallowing by oral tube**	During the swallowing execution, the primary sensorimotor cortex α and β activity is left-sided during [−400:+200 ms] then is symetric during [+200 ms:+400 ms] and last, right-sided during [+400:+600 ms] (in reference to muscle activation called M1 in the study).	BA1,2,3 BA4
**Adaptive physiology**				
Teismann 2007 [29]	**To study cortical activity during self-paced volitional swallowing with and without topical oropharyngeal anesthesia with MEG** To evaluate the impact of sensory input in healthy subjects.	**Task 1: Self-paced volitional 10 mL/min water swallowing by oral tube** **Task 2: Self-paced volitional 10 mL/min water swallowing by oral tube after pharyngeal anesthesia**	During volitional swallowing, β activity is bilateral within the primary sensorimotor cortex and maximum at 300 ms. Peripheral sensory suppression reduces the cortical responses, most predominantly on the left side (−35%, *p* < 0.05) VS the right side (−28%, *p* < 0.05) without significant lateralization	BA1,2,3 BA4
Teismann 2009 [31]	**To study cortical activity during self-paced volitional swallowing with and without preceding thermal tactile oral stimulation**	**Condition 1: Self-paced volitional 10 mL/min water swallowing by oral tube** **Condition 2: Self-paced volitional 10 mL/min water swallowing by oral tube after TTOS**	In the control condition, the primary sensorimotor cortex α and β activity is stable during [−400:0 ms] then is left-sided during [0:200 ms] and right-sided during [200:600 ms]. Cold stimulation (TTOS) improves the left α and β activity (*p* < 0.05) during the whole execution sequence with a left lateralization through [−400 ms:+600 ms]. This suggests the volitional (Oral) phase seems more left-sided and the reflexive phase (pharyngo-oesophageal) seems more right-sided.	BA1,2,3 BA4
Teismann 2010 [34]	**To examine with whole-head MEG and compare changes in cortical swallowing processing in young versus elderly subjects**	**Condition 1: Young volunteers—Self-paced volitional 10 mL/min water swallowing by oral tube** **Condition 2: Elder volunteers—Self-paced volitional 10 mL/min water swallowing by oral tube**	In elders, broader and stronger bilateral (*p* < 0.05) activation during preparation and execution in comparison to classical results in young subjects (BA4,3,2,1 α and β ERD bilateral symmetrical activity]	BA1,2,3 BA4
**Adaptive physiology**				
Suntrup 2013 [38]	To evaluate the effect of tDCS on the swallowing network activity by applying MEG. **To gain insight into the underlying mechanism of action and to link neuroplastic with behavioral changes in swallowing.**	**Task 1: Pre-Tdcs—Visually cued simple saliva swallowing—“Simple swallow task”** Task 1: Post-tDCS—Visually cued simple saliva swallowing—“Simple swallow task” **Task 2: Pre-tDCS—Visually cued fast saliva swallowing—“Fast swallow task”** Task 2: Post-tDCS—Visually cued fast saliva swallowing—“Fast swallow task” **Task 3: Pre-tDCS −150 ms time window-targeted saliva swallowing—“Challenged swallow task”** Task 3: Post-tDCS—150 ms time window-targeted saliva swallowing—“Challenged swallow task”	In control condition, activity similar to previous reports. The fast swallow task after tDCS increases the pericentral activity in all bands (*p* = 0.006). The challenged task after tDCS increases both pericentral and premotor (PMC and SMA) activity in all bands, and also the parieto-occipital α activity (*p* = 0.007)	BA1,2,3 BA4 BA6
Suntrup 2015 [40]	**To contribute further knowledge on the cortical topography and frequency–specificity of activation pattern changes during the act of swallowing by taking advantage of MEG.** **To analyze the complete act of swallowing instead using a method allowing to explore the stimulation-induced alterations in the cortical large-scale oscillatory swallowing network beyond the pharyngeal motor cortex.**	**Condition 1: Before pharyngeal electrical or sham stimulation—Self-paced volitional 10 mL/min water swallowing by oral tube** **Condition 2: Immediately (about 6 min) after pharyngeal electrical stimulation—Self-paced volitional 10 mL/min water swallowing by oral tube** **Condition 3: Immediately (6 min) after sham stimulation—Self-paced volitional 10 mL/min water swallowing by oral tube** **Condition 4: 40–55 min after pharyngeal electrical or sham stimulation—Self-paced volitional 10 mL/min water swallowing by oral tube**	Control conditions (n°1, 3 and 4) displays similar results to previous reports. Right decrement during PES	BA1,2,3 BA4 BA6 BA9, 10, 45 BA44 BA13 BA40 BA43
**Adaptive physiology**				
Muhle 2021 [42]	**To investigate whether anodal tDCS (transcranial Direct Current Stimulation) and PES (Pharyngeal Electrical Stimulation) can reverse the effects of experimentally induced pharyngeal hypesthesia on the cortical swallowing network using MEG, using a “virtual lesion model” based on local anesthesia**	**Task 1: Baseline post local anesthesia-Self-paced volitional 10 mL/min water swallowing by oral tube after local pharyngeal anesthesia** **Task1: A-After tDCS—Self-paced volitional 10 mL/min water swallowing by oral tube after local pharyngeal anesthesia** **Task 1: C–After Pharyngeal Electrical Stimulation—Self-paced volitional 10 mL/min water swallowing by oral tube after local pharyngeal anesthesia** Task 2: Baseline post local anesthesia-Pneumatic pharyngeal stimulation for 15 min through transnasal catheter Task 2: B–After tDCS—Pneumatic pharyngeal stimulation for 15 min through transnasal catheter Task 2: D–After PES—Pneumatic pharyngeal stimulation for 15 min through transnasal catheter	After pharyngeal anesthesia, beta, alpha and theta ERD are seen in pericentral cortex with maximum activity in BA6R (*p* = 0.047) PES had a positive treatment effect on cortical activity (*p* = 0.01) whereas tDCS had not. PES might be useful for peripheral damage of the swallowing system, whereas tDCS might be limited to central damage (*p* > 0.05) In their peripheral sensory lesion model of dysphagia, PES as a peripheral stimulation method was able to revert the detrimental effects of reduced sensory input on central swallowing processing, whereas tDCS as a central neuromodulation technique was not. Results may have implications for therapeutic decisions depending on the nature of dysphagia in the clinical context.	BA1,2,3 BA4 BA6
Suntrup-Krueger 2021 [43]	**To comprehensively investigate the effect of oral application of a capsaicin-containing red pepper sauce suspension on the biomechanics and neurophysiology of swallowing.** To gather further information on the feasibility of capsaicin treatment for dysphagia potential desensitization due to overstimulation was evaluated and the duration and intensity of the effect were assessed by monitoring salivary SP level over time.	**Condition 1/Task 1: 5 min preconditioning with pure water + 15 min Self-paced 10 mL/min pure water swallowing by oral tube** **Condition 1/Task 2: 5 min preconditioning with pure water + Challenged 10 mL/min pure water swallowing by oral tube** **Condition 2/Task 1: 5 min preconditioning with capsaicinoids + 15 min Self-paced 10 mL/min pure water swallowing by oral tube** **Condition 2/Task2: 5 min preconditioning with capsaicinoids + challenged 10 mL/min pure water swallowing by oral tube** **Condition 3/Task1: 5 min preconditioning with capsaicinoids + 15 min Self-paced 10 mL/min capsaicinoids swallowing by oral tube** **Condition 3/Task2: 5 min preconditioning with capsaicinoids + challenged capsaicinoids 10 mL/min swallowing by oral tube**	In control condition (Condition 1/Task1), activity was similar to previous reports. Challenging conditions (Task 2) increased α and β activity in parieto-occipital cortex Capsaicinoids had no effect on cortical MEG but had a direct peripheral effect	BA1,2,3 BA4 BA6
**Pathology**				
Teismann 2008 [30]	**To study the clinical and neurofunctional changes in swallowing performance and central swallowing processing during remission from botulism intoxication.**	**Condition 1: Self-paced volitional 10 mL/min water swallowing by oral tube—15 healthy subjects** **Condition 2: Slow self-paced volitional 10 mL/min water swallowing by oral tube—1 healthy subject** **Condition 3: Self-paced volitional 10 mL/min water swallowing by oral tube—1 botulism subject—** **Day 20** **Condition 4: Self-paced volitional 10 mL/min water swallowing by oral tube—1 botulism subject—** **Day 25**	During volitional swallowing, β activity is bilateral within the primary sensorimotor cortex with a max ERD in BA6. γ activity of the insula is linked to the swallowing frequency. Volitional reduction in the swallowing frequency reduces the activity in insula and shift the pericentral activity to the right. In the botulism patient, specific β activity of the pericentral cortex disappeared and a specific BA7 activity is observed. However, from a pathophysiological point of view, it is hard to conclude whether the modifications of the cortical activity (loss of activity) are due to Botulism cerebral lesions themselves or to swallowing frequency reduction.	BA1,2,3 BA4 BA6 BA7 (pat.) BA13
Dziewas 2009 [33]	**To investigate the cortical topography of volitional swallowing in patients with Kenedy disease**	**Condition 1: Kenedy disease—-paced volitional 10 mL/min water swallowing by oral tube** **Condition 2: Healthy controls—Self-paced volitional 10 mL/min water swallowing by oral tube**	In controls, during swallowing, the β activity of the primary motor cortex is focused in a small area and is left-sided during the preparation then more symmetric during execution (*p* < 0.05). In patients, during swallowing, the β activity of the primary motor cortex is stronger and extended to PFC and posterior parietal cortex and globally right-sided during preparation and execution (*p* < 0.05)	BA1,2,3 BA4 BA6 BA7 (pat.) BA9,10
Teismann 2011 [35]	**To examine cortical swallowing processing in patients in this early subacute phase after stroke of the cerebrum or the brainstem and focusing on the side of the lesion.**	**Condition 1: Healthy controls—Self-paced volitional 10 mL/min water swallowing by oral tube** **Condition 2: Hemispheric stroke without dysphagia—Self-paced volitional 10 mL/min water swallowing by oral tube** **Condition 3: Hemispheric stroke with dysphagia—Self-paced volitional 10 mL/min water swallowing by oral tube** **Condition 4: Brainstem stroke—Self-paced volitional 10 mL/min water swallowing by oral tube**	In controls, activation is similar to previous studies with β ERD in [BA4,6 +1,2,3,5,7] (*p* < 0.05) In case of hemispheric stroke, higher activity of the DLPFC and insula. The presence of dysphagia modifies the results. Hemispheric stroke with dysphagia shows a reduction in ipsilateral pericentral activity with no contralateral activity whereas non dysphagic subject shows bilateral activity similar to controls. Brainstem stroke patients shows a right lateralization of their pericentral activity.	BA1,2,3 BA4 BA6 BA45,46,47
**Pathology**				
Teismann 2011 [36]	**To study cortical activity during self-paced volitional swallowing on fourteen patients suffering from sporadic ALS with bulbar onset with MEG.**	**Condition 1: Healthy controls—Self-paced volitional 10 mL/min water swallowing by oral tube** **Condition 2: Mildly dysphagic patients—Self-paced volitional 10 mL/min water swallowing by oral tube** **Condition 3: Severely dysphagic patients—Self-paced volitional 10 mL/min water swallowing by oral tube**	Healthy controls display similar results to previous reports. In ALS, the more dysphagic, the more right lateralized is the activity, with a global reduction in the pericentral activity in comparison to controls (*p* < 0.05). No local extension of activity.	BA1,2,3 BA4 BA6
Suntrup 2013 [37]	**To evaluate differences in swallow-related cortical activation in dysphagic versus non-dysphagic patients with Parkinson’s disease and healthy control subjects using an established swallow paradigm**	**Task: Controls—Self-paced volitional 10 mL/min water swallowing by oral tube** **Task: Non-dysphagic Parkinson’s patients—Self-paced volitional 10 mL/min water swallowing by oral tube** **Task: Dysphagic Parkinson’s patients—Self-paced volitional 10 mL/min water swallowing by oral tube**	In all 3 groups: bilateral pericentral sensorimotor activation In patients, a strong decrease in activation was found (*p* < 0.05) In non-dysphagic patients: shift of peak activation toward lateral motor, premotor and parietal cortices, reduced and delayed SMA activity (*p* < 0.01) In dysphagic patients, reduced activation restricted to the sensorimotor areas (*p* < 0.05).	BA1,2,3 BA4 BA6 BA40 BA43
Suntrup 2014 [39]	**To investigate cortical swallow-related activation in patients diagnosed with functional dysphagia by means of MEG** **To determine whether functional dysphagia is associated with alterations in cortical swallowing processing.**	**Condition 1: Healthy controls—Self-paced volitional 10 mL/min water swallowing by oral tube** **Condition 2: Functional dysphagic subjects—Self-paced volitional 10 mL/min water swallowing by oral tube**	Healthy controls display similar results to previous reports. In functional dysphagic patients, the pericentral activity is reduced and right lateralized (LI = −0.5050) with specific activity of the right SMA, right insula, right DLPFC and right inferolateral parietal lobe. Pericentral activity in healthy subjects is more rostro-medial and in functional dysphagic patient, is more caudo-lateral. Right lateralization in patients	BA1,2,3 BA4 BA6 BA9,45 BA44 BA13 BA40 BA43
**Pathology**				
Suntrup-Krueger 2018 [41]	To contribute robust evidence to the value of tDCS in dysphagia rehabilitation and overcome some limitations of previous studies. To evaluate the efficacy of a patho-physiologically reasonable tDCS protocol to improve stroke-related oropharyngeal dysphagia, conducting a randomized controlled trial (RCT) in a sufficiently large patient sample with objective clinical outcome measures alongside functional neuroimaging. To identify predictors of treatment success, which they hypothesized to be patient-related (age, lesion location/size, stroke and OD severity) and/or treatment-related (timing, tDCS + training vs tDCS alone).	**Condition 1: Sham group—Self-paced volitional 10 mL/min water swallowing by oral tube** **Condition 2: tDCS group—Self-paced volitional 10 mL/min water swallowing by oral tube**	Control conditions (n°1) displays similar results to previous reports.	BA1,2,3 BA4 BA6 BA45 BA23,31 BA40

BA: Brodmann area; NF: Neurofeedback; Pat.: Results specific to pathological subjects. R/L: Right/Lefta. ^a^ Correspondence between 10-10 system, Brodman areas and anatomical gyri were performed according to the article data but also Scrivener and Reader 2022 [56], Okamoto 2004 [57] and http://bioimagesuite.com (based on Lacadie et al. 2008 [58], accessed on 22 April 2022) when coordinates were available.

## Data Availability

Not applicable.

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
