# Peer review of "Can Swallowing Cerebral Neurophysiology Be Evaluated during Ecological Food Intake Conditions? A Systematic Literature Review"

_jcm, 2022, doi:10.3390/jcm11185480_

Round 1
Reviewer 1 Report
Thank you for giving me the opportunity to review this manuscript. This is an interesting systematic review on functional imaging studies – other than fMRI or PET – that investigated cerebral activation during swallowing in an ecological bolus intake condition. The field has grown distincly during the last couple of years but a systematic review on the specific topic has not been provided so far which makes this manuscript a relevant contribution to the field. Functional imaging in the field of swallowing research not only lead to a better understanding of the underlying (patho-)physiology but also gives opportunity to optimize targeting for neurostimulatory devices. The methods used for this review as well as the structure and language are sound. There are only a few minor concerns that I think need be addressed:
- The formatting of schemes and supplementary tables needs to be revised. The way it is presented now makes it difficult to follow the content.
- The derivation of the research question (introduction section) could be a little more detailed. The field that is dealt with is a bit of a niche within swallowing research and a more extensive explanation why this topic was chosen for this systematic review may lead to better acceptance outside of the field of dysphagia experts.
- There are a few minor spelling issues that should be rechecked, e.g. paragraph 1 in the introduction: retroambiguus or 4.1.1 on the one hand rather than one the one hand.
- The aspect that brainstem activity cannot be detected by EEG, MEG or fNIRS is obvious and stated in the manuscript. However, it the implication of this method-related limitation may be discussed more extensively in the discussion section.
Author Response
Dear Reviewer,
We would like to thank you for your helpful and relevant comments. Here you will find the answers to your suggestions for corrections.
- The formatting of schemes and supplementary tables needs to be revised. The way it is presented now makes it difficult to follow the content.
-
In our original submission, there were no “schemes” and only "supplementary tables". We added to the manuscript short tables as “schemes” to improve reading and kept the complete tables as supplementary material. We hope it will help to follow the content without losing information.
- The derivation of the research question (introduction section) could be a little more detailed. The field that is dealt with is a bit of a niche within swallowing research and a more extensive explanation why this topic was chosen for this systematic review may lead to better acceptance outside of the field of dysphagia experts.
-
We added some examples of subjects of interest for whom fMRI is not well suited, to better explain the research of new “ecological” techniques.
- There are a few minor spelling issues that should be rechecked, e.g. paragraph 1 in the introduction: retroambiguus or 4.1.1 on the one hand rather than one the one hand.
-
We corrected those typos and some other we found
- -The aspect that brainstem activity cannot be detected by EEG, MEG or fNIRS is obvious and stated in the manuscript. However, it the implication of this method-related limitation may be discussed more extensively in the discussion section.
-
We added some points discussing this matter in the discussion section
Reviewer 2 Report
There are a lot of tables in the paper, the tables are very extensive. You can reedit and make it more visual friendly or put there in supplemenary materials.
Author Response
Dear Reviewer,
We would like to thank you for your helpful and relevant comments. Here you will find the answers to your suggestions for corrections.
- There are a lot of tables in the paper, the tables are very extensive. You can reedit and make it more visual friendly or put there in supplementary materials.
- In our original submission, there were no “schemes” and only "supplementary tables". We added to the manuscript short tables as “schemes” to improve reading and kept the complete tables as supplementary material. We hope it will help to follow the content without losing information.
Reviewer 3 Report
Dear Editor, thank you for the opportunity to revise this paper.
The topic is of interest, and contribute to the improvement of the understanding of neuroanatomical correlates of a complex function such as swallowing.
The procedure of the review is correct to me.
I only have a minor suggestion, which is to try to shorten or simplify to the reader scheme 1-2-3, and supplementary tables. Also, the vertical orientation of the table doesn't help the reader.
Author Response
Dear Reviewer,
We would like to thank you for your helpful and relevant comments. Here you will find the answers to your suggestions for corrections.
- I only have a minor suggestion, which is to try to shorten or simplify to the reader scheme 1-2-3, and supplementary tables. Also, the vertical orientation of the table doesn't help the reader.
- In our original submission, there were no “schemes” and only "supplementary tables". We added to the manuscript short tables as “schemes” to improve reading and kept the complete tables as supplementary material. We hope it will help to follow the content without losing information.